# Serum Interleukin-8 in Patients with Different Origin of Intra-Abdominal Infections in Perioperative Period

**DOI:** 10.3390/medsci7090094

**Published:** 2019-09-08

**Authors:** Artem Riga, Valeriy Boyko, Yuriy Grirorov

**Affiliations:** Surgery N 1 department, Kharkiv National Medical University Kharkiv, 61022, Ukraine (V.B.) (Y.G.)

**Keywords:** intra-abdominal infections, serum interleukin-8, perioperative period

## Abstract

Intra-abdominal infections (IAI) are associated with high levels of pro-inflammatory serum IL-8 and poor outcomes, but data on IL-8 levels in various inflammatory reactions are contradictory. A better understanding of the diagnostic role of IL-8 is important, since the clinical relevance remains unclear. **Methods:** That was a single-center observational longitudinal cross-sectional study included 56 patients with various origins of intra-abdominal infections: 24 patients with postoperative abscesses, 12 patients with primary intra-abdominal abscesses, and 20 patients with diffuse peritoneal collection. Perioperative serum concentrations of interleukin-8 IL-8 were investigated at the day before surgery, on the 2nd–3rd day, and on the 5th–7th day after surgery. The hypothesis suggested that there was a difference in serum IL-8 in patients with IAI of different origin in the perioperative period. **Results:** The study showed that the level of serum IL-8 in patients with intra-abdominal infections of different origins is lower in comparison with healthy individuals. Despite the fact that we did not detect any statistically significant differences in the level of IL-8 in serum in IAI of different origin in the perioperative period, its lowest index was observed in the patients with postoperative abscesses on the 5th–7th days after surgical intervention. The levels of serum IL-8 ≤49.71 pg/mL and ≤48.88 pg/mL may serve as diagnostic markers for primary and postoperative abscesses with significant sensitivity and specificity. **Conclusions:** Our results differ from previous studies that showed high serum IL-8. High-quality clinical trials are needed to better comprehend the role of inflammatory mediators in IAI with different origin.

## 1. Introduction

The immune system is an important regulator of the response to intra-abdominal infections (IAI). Moreover, even different surgical approaches, such as laparotomy and laparoscopy, in different ways influence the immune response and development of intra-abdominal sepsis that was shown in experimental models [1,2]. The immune system activates and instructs adaptive immune responses, regulates inflammation, and mediates immune homeostasis (the balance between opposing pro-inflammatory and anti-inflammatory processes) [3]. Interleukin-8 (IL-8) is a pro-inflammatory multifunctional cytokine with different physiological functions, and the role of IL-8 in IAI was still missing from the big picture [4]. It modulates local and systemic acute and chronic inflammatory reactions [5]. Some studies demonstrate correlation of the IL-8 level with sepsis and postoperative trauma [6,7]. Investigation of normal IL-8 in a healthy individuals’ response after intravenous introduction of endotoxin showed a maximal peak after 2 h and duration of 12 h [8]. 

IL-8 has been studied as a biomarker for many clinical conditions and is currently used by various medical subspecialties, either for rapid diagnosis or as a predictor of prognosis. IL-8 was studied in various hematological and oncological diseases, urinary tract diseases, autoimmune diseases, and sepsis [9,10,11,12,13,14].

A unified consensus and understanding of the role of serum IL-8 as a valid biomarker for early diagnosis and/or a biomarker for the prognosis of the different diseases, especially local infectious processes, is still being studied. For example, Hikmet Kocak demonstrated that IL-8 was not the best marker for early diagnosis of bladder cancer, but might serve as a predictor of prognosis [15]. There is still not enough data on the IL-8 as a diagnostic biomarker in patients with IAI, especially in postoperative intra-abdominal abscesses.

IAI are still regarded as an important cause of morbidity and mortality worldwide, despite antibiotics application [16,17]. These infections are caused by Gram-negative, aerobic, and anaerobic mandatory and often mixed bacterial flora [18]. Early diagnosis of the IAI is important to assess the severity and improve the prognosis of disease [19]. The factors influencing progression of IAI include advanced age, malnutrition, pre-existing diseases, use of immunosuppressants, prolonged peritonitis, virulence of the pathogenic microorganism, and actual immune response [20]. However, the clinical relevance of these findings is not clear. Many clinical conditions generally require expensive and time-consuming investigations for their diagnosis. There is therefore a general need for exploring noninvasive markers in clinical medicine.

**Objective:** investigation of pro-inflammatory IL-8 serum concentration in patients with different origin of IAI in the perioperative period.

**Hypothesis:** There is a difference in serum IL-8 in patients with intra-abdominal infection of different origin in the perioperative period.

## 2. Methods

This was a single-center observational longitudinal cross-sectional study. Clinical specimens were obtained from 56 patients with different IAI attended to at the Hospital of State Enterprise “’Institute of General and Emergency Surgery named after V.T.Zaytsev’ National Academy of Medical Science NAMS of Ukraine” in 2016–2018. This study was authorized by the Ethics Committee of the Kharkiv National Medical University (Report No 6 from 5 October 2016). Inclusion criteria: Patients with primary and secondary IAI. The distribution of patients with IAI was as follows: 24 patients with postoperative abscesses (POA), 12 patients with primary intra-abdominal abscesses (PIAA), and 20 patients with diffuse peritoneal collection (PC) due to IAI only. Exclusion criteria: Patients who did not present IAI and those who did not agreed to participate in this study. For immunological analysis, 5 mL blood was obtained—by vacuum sterile disposable tubes (Vacutainer) at the day before surgical intervention, on the 2nd–3rd day, and on the 5th–7th day after surgical intervention and from 31 healthy volunteers (Control group). Serum was centrifuged at 800 g for 10 min at 4 °C and was being stored at −20 °C for three month, as recommended by the manufacturer. Extracellular IL-8 in the serum was assessed with the VectorBEST test system (Kyiv, Ukraine) cytokines kit, according to manufacturer’s instructions, in immune enzyme analyzer “LabLine-90” (Austria). Statistical analysis was performed with the MedCalc version 14.8-© 1993-2014 MedCalc Software bvba (Acacialaan 22 B-8400 Ostend, Belgium). Descriptive analysis: Kruskal–Wallis ANOVA test (KW), Kaplan–Meier survival analysis, and Receiver Operating Characteristic (ROC) curve analysis. The difference in distribution of the serum IL-8 among polymorphic variant groups (*n* ≥ 3) was compared by two-way Kruskal–Wallis ANOVA test (KW) of variance with power calculation. The value was considered significant if the power was more than 0.8. ROC curves were constructed by calculating the sensitivities (true positive rate) and specificities (false positive rate) of all assays at several cut-off points with calculating area under the curve (AUC). For all statistical methods, *p* < 0.05 was considered statistically significant.

## 3. Results

Among 56 patients with IAI of different origin engaged from the hospital with the average age 56 years old (ranging from 19 to 83 years), 26 were men. The common comorbidities were systemic hypertension and diabetes mellitus type 2 (Table 1). About half (26/56) reported previous recent surgery due to IAI. The similar amount (26/56) of the patients underwent surgery using minimally invasive technologies. All patients took antibacterial therapy, according to hospital local protocol.

Among 31 healthy individuals of the Control group there were 18 males and 13 females. The median age of them was 57 (minimum, 32; maximum, 77) years (KW test H = 3.0, *p* = 0.2203). There was neither pregnancy nor complaints at the time of blood collection, nor previous surgical interventions, nor medication within the last 72 h, nor chronic diseases.

The Gram-positive infection developed in 25 (48.6%) patients. Gram-negative infection developed in 22 patients (39.2%). In 2 from 56 patients (3.6%) bacterial flora was not detected. It is explained by the fact that Gram-negative flora is found significantly more often in patients with postoperative abscesses (21/23) (*p* = 0.0004). There was no possibility to isolate flora in one patient with postoperative abscess.

The day before surgical intervention, the total number of leukocytes was statistically significantly lower in patients with postoperative abscesses and primary intra-abdominal abscesses compared with the patients with peritoneal collection, although neutrophils (%) did not differ.

We found that individuals with different origin of IAI displayed different lengths of hospitalization (KW test H = 6.8, *p* = 0.0320). Kaplan–Meier analysis demonstrated that patients with postoperative abscesses have had more prolonged hospitalization (>40 days), despite the use of minimally invasive technology in 18 (75%) of them. 

In order to gain insight about the inflammatory processes associated with IAI, we evaluated the serum IL-8 (*n* = 56) of these patients before and during 7 days after surgical intervention. The lowest index detected in the Control group (healthy individuals) was 16.67 pg/mL, the highest—96.3 pg/mL for IL-8; median serum concentration was 72.07 pg/mL. A difference was found between distributions of serum IL-8 in patients with postoperative abscesses the day before surgical intervention and on the 2nd–3rd day of the postoperative period, with a significant decrease on 5th–7th days compared with the healthy individuals (Figure 1) (KW test H = 7.1, *p* = 0.0285). There was no significant difference of serum IL-8 distribution in patients with primary intra-abdominal abscesses or peritoneal collection in the perioperative period (Figure 2 and Figure 3) (KW test H = 0.4, *p* = 0.7969 and H = 2.2, *p* = 0.3296).

As it was expected, the levels of IL-8 in the serum did not differ in individuals with different origin of IAI. For example, the median serum concentration of IL-8 was 37.68 pg/mL in patients with postoperative abscesses, 27.21 pg/mL in patients with primary intra-abdominal abscesses, and 31.62 pg/mL in patients with peritoneal collection due to IAI the day before surgical intervention (KW test H = 2.8, *p* = 0.2381). The median concentration of IL-8 was 38.33 pg/mL in patients with postoperative abscesses, 26.28 pg/mL in patients with primary intra-abdominal abscesses, and 28.47 pg/mL in patients with peritoneal collection on the 2nd–3rd day after surgical intervention (KW test H = 3.5, *p* = 0.1659). The median concentration of IL-8 was 21.85 pg/mL in patients with postoperative abscesses, 22.38 pg/mL in patients with primary intra-abdominal abscesses, and 29.11 pg/mL in patients with peritoneal collection on the 5th–7th day after surgical intervention (KW test H = 0.5, *p* = 0.7417). The significant difference was found when comparing the concentration IL-8 in serum in patients with IAI and healthy persons (KW test H = 35.9, *p* < 0.0001).

The distribution of the serum IL-8 level in patients with different origin of IAI before surgical intervention, on the 2nd–3rd and the 5th–7th days of postsurgical intervention see in Appendix A (Appendix A).

The prevalence of a disease may be different in different clinical settings. For our cohort we evaluated sensitivity and specificity of serum IL-8 for IAI in the perioperative period for applies (Table 2).

Of note, the lowest concentration with significant sensitivity and specificity of serum IL-8 was for primary and secondary intra-abdominal abscesses (*p* < 0.0001).

The curves of ROC analysis for Table 2 see in Appendix A (Appendix A).

We checked our results in two-way ANOVA test where independent factors for IL-8 were time and IAI of different origin (see in Appendix A
Appendix A and Appendix A).

## 4. Discussion

Published studies have associated IAI with a prolonged morbidity and significant mortality rate [16,17,18,19,20]. This situation can be associated with some reasons, for example, inadequate empirical antibacterial treatment and poor inflectional control that leads to increasing bacterial resistance [21,22]. Despite increased knowledge about epidemiology and antibiotic therapy, applying minimally invasive technology, our results were similar to those obtained by Vallejo M et al. [23]. Data from Solomkin J.S. et al. show that IAI is still one of the major challenges to the health system [24].

IAI are associated with systemic inflammatory processes. In order to find out how the inflammation was modulated by the infection origin, we measured pro-inflammatory cytokine IL-8 in the serum of patients with IAI in the perioperative period. We have also investigated the serum level of IL-8 in 31 healthy individuals. Until now, there is different data regarding the levels of interleukins in intra-abdominal infections. Some publications show high serum levels of IL-8 associated with severe infection and poor outcomes [25,26]. Moreover, no correlation was found between the content of cytokines in serum and in the focus of inflammation [2]. Our results with low serum IL-8 level in the perioperative period in patients with IAI of different origin coincide with the data received by Zhengwen Xiao et al. and João Fernando Gonçalves Ferreira et al. which proved that the greatest concentration of IL-8 is contained in the inflammatory focus compared with the serum [17,18].

We showed that the levels of IL-8 in the serum were not different in individuals with different origin of IAI. On the other hand, the lowest concentration of serum IL-8 was in patients with postoperative abscesses at the 5th–7th days after surgical intervention. Currently, postoperative infection is one of the leading causes of sepsis and mortality in patients with IAI. The reason is the nosocomial flora, resistant to antibiotic therapy. The multicenter observational study so-called as WISS (World Society of Emergency Surgery of complicated intra-abdominal infections Score Study), with 4553 patients from 132 hospitals worldwide during 15 October 2014–2015 February 2015, dedicated to validation of patients with complicated IAI, found out that the postoperative infection was in 387 (8.5%) patients and occupied the third position among all source of IAI [27].

We suppose that nosocomial infections exhibit a completely different inflammatory response compared with community-acquired infections, although immunological reactivity should be considered an individual process, which depends on the microbial agent [28].

The next special issue for discussion is Diabetes mellitus, because it is a risk factor for the development of IAI [29]. Chronic hyperglycemia contributes to an increase in the risk of gastrointestinal infections [30]. Concerning cellular innate immunity, most studies show decreased functions of polymorphonuclear cells and monocytes/macrophages. Another mechanism which can lead to the increased prevalence of infections in diabetic patients is an increased adherence of microorganisms to diabetic compared with nondiabetic cells [31]. But the data concerning the level of cytokines are contradictory. So, the study conducted by Cimini F.A. et al. demonstrated that serum IL-8 was increased in diabetic patients [32]. Some investigators claim that the differences in the risk factors for infection between diabetic and nondiabetic patients result either from noncontrolled studies or from biased studies However, most researchers conclude that there is clinical evidence pointing to the higher prevalence of infectious diseases among individuals with Diabetes mellitus [29,33].

Accordingly, Barnett et al. also observed that nosocomial-acquired infections are more severe, requiring longer hospitalization and showing higher death rates in risk-group patients [34].

Pro-inflammatory serum IL-8 was measured only at the peak of disease (peak of inflammation, surgical trauma); they were not continuously monitored at different points of time. Thus, the peak levels of these cytokines were unclear. Measurement of cytokines should be further investigated as a more sensitive determinant of intra-abdominal inflammatory response. 

There were some inherent limitations associated with this study; firstly, sample size. Our model was based on prospective single-center cross-sectional study and was limited by the time and number of patients in the East Ukrainian population. Secondly, there were very few prior researches and gaps in the studies relevant to serum IL-8 in patients with different origin of IAI, which influenced the methodology of our study. Our study was limited by the early perioperative period. There were confounders such as Diabetes mellitus that could influence the serum levels of this pro-inflammatory biomarker [32,33,35]. We were unable to assess whether there was a joint effect of Diabetes mellitus and IAI on cytokine production, which may undermine the strength of the research. As the result, ROC analysis detecting all types of IAI might have misclassified POA as PIAA or PC, which would bias the results of our overall and subgroup analysis toward the null hypothesis. Third, etiological effects of microbial agents on changes in serum IL-8 were not estimated due to diversity and combination. Further investigations of our and different population of pro-inflammatory markers are needed to ascertain their relevance both as early predictor as well as diagnostic cut-offs for IAI. Large-scale studies are needed for the usefulness and effectiveness of IL-8 as a biomarker, estimated from meta-analyses with validity confirmatory analysis, with people living in other territories.

## 5. Conclusions

The study showed that the level of serum IL-8 in patients with intra-abdominal infections of different origin is lower in comparison with healthy individuals. Its results differ from previous studies that indicated an increase of IL-8. Despite the fact that we did not detect any statistically significant differences in the level of IL-8 in serum in the IAI of different origin in the postoperative period, its lowest index was observed in the patients with postoperative abscesses on the 5th–7th days after surgical intervention compared with the Control group. The levels of serum IL-8 ≤49.71 pg/mL and ≤48.88 pg/mL may serve as diagnostic markers for primary and postoperative abscesses with significant sensitivity and specificity. The relationship between the cytokine response and the specific nosocomial bacterial agent, the mechanisms of localization, and generalization of the infection process should be further studied. High-quality clinical trials are needed to better comprehend the role of inflammatory mediators in IAI with different origin.

## Figures and Tables

**Figure 1 medsci-07-00094-f001:**
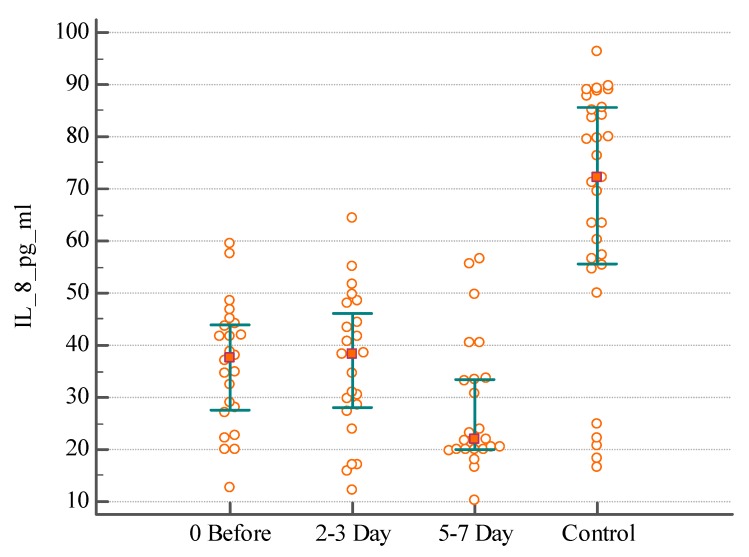
The distribution of serum IL-8 level in patients with postoperative abscesses before, and on the 2nd–3rd and 5th–7th days after surgical intervention.

**Figure 2 medsci-07-00094-f002:**
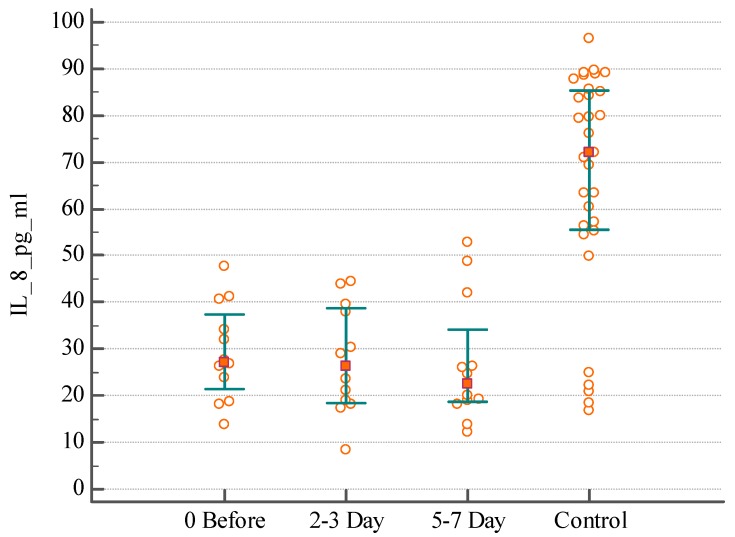
The distribution of serum IL-8 level in patients with primary intra-abdominal abscesses before, and on the 2nd–3rd and 5th–7th days after surgical intervention.

**Figure 3 medsci-07-00094-f003:**
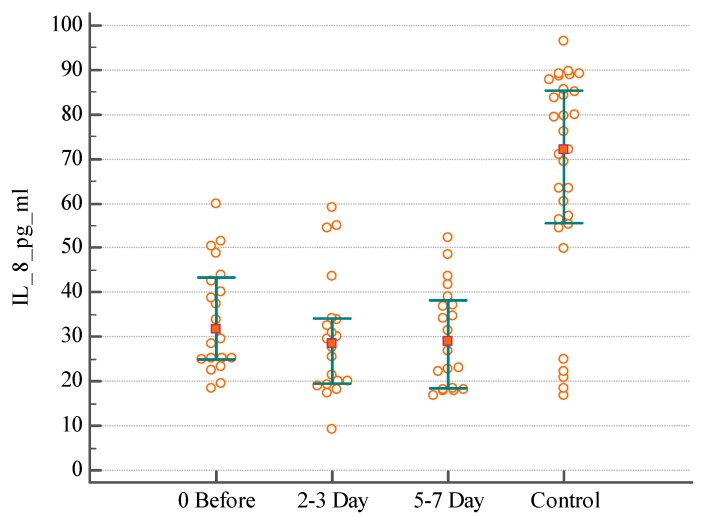
The distribution of serum IL-8 level in patients with diffuse peritoneal collection due to intra-abdominal infections before, and on the 2nd–3rd and 5th–7th days after surgical intervention.

**Table 1 medsci-07-00094-t001:** Demographic and clinical variables of patients with different origin of intra-abdominal infections (IAI).

Data	POA *n* = 24	PIAA *n* = 12	PC *n* = 20	Total *n* = 56
Age, years (Median, (min; max))	58 (30; 78)	63 (52; 79)	48 (19; 83)	56 (19; 83)
Male	10	4	12	26
Female	14	8	8	30
Diabetes mellitus type 2	2	3	2	7
Systemic arterial hypertension	2	1	2	7
Previous abdominal surgical interventions	24	1	1	26
Duration in hospital, days (Median, (min; max))	14 (7; 98)	18 (12; 34)	11 (6; 33)	13 (6; 98)
Routine surgical intervention (laparotomy)	6	4	20	30
Minimally invasive surgical intervention (abscess punction)	18	8	0	26
Abscess or peritoneum culture				
Gram positive	3	9	13	25
*Enterococcus fecalis*	1	3	6	10
*Staphilococcus aureus*	-	4	6	10
*Staphilococcus epidermidis*	1	1	3	5
Gram negative	19	1	2	22
*Escherihia coli*	4	3	11	18
*Enterobacter aerogenes*	2	-	1	3
*Enterobacter* spp.	3	-	-	3
*Pseudomonas aeruginosa*	1	-	2	3
*Edwardsiella tarda*	1	-	-	1
*Klebsiella* spp.	1	-	-	1
Poly	2	2	3	7

**Table 2 medsci-07-00094-t002:** ROC analysis result for serum IL-8 in patients with IAI of different origin.

IAI	AUC	Concentration, pg/mL	*p*	Sensitivity (95% Confidential Interval)	Specificity 95% Confidential Interval)	Youden Index J
Post-operative abscesses	0.860	≤49.71	<0.0001	90.28 (81.0–96.0)	83.87 (66.3–94.5)	0.7415
Primary intra-abdominal abscesses	0.888	≤48.88	<0.0001	97.22 (85.5–99.9)	83.87 (66.3–94.5)	0.8109
Diffuse peritoneal collection	0.864	≤54.35	<0.0001	95.00 (86.1–99.0)	80.65 (62.5–92.5)	0.7565
Total	0.850	≤51.52	<0.0001	94.64 (85.1–98.9)	80.65 (62.5–92.5)	0.7529

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
