# Peer review of "Serum Interleukin-8 in Patients with Different Origin of Intra-Abdominal Infections in Perioperative Period"

_medsci, 2019, doi:10.3390/medsci7090094_

Round 1

Reviewer 1 Report

The authors presented a study of serum IL-8 levels in patients with IAI and they found that IL-8 level in patients with different level of IAL are important for diagnosis in IAI. There are some minor points raised by this reviewer,

1, The introduction could be improved. At least, the authors could add a short paragraph in the introduction section to discuss the current understanding of IL-8 in different disease, which will be helpful to broader readers.

2, In Figure 1, 2 and 3, labels for the y-axis are missing.

3, Previously, IL-8 has been well known to induce neutrophils migration and neutrophils are essential for protection against the infection. Since the authors found IL-8 level is reduced in IAI patients, a discussion on the consequence of lower IL-8 level will provide an deeper insight into the IL-8 mediated immunity in IAI diseases.

Author Response

Dear Reviewer,

The authors made all changes and comments. They are attached to the document. Thank you for the opportunity to improve our manuscript.

Best wishes.

Authors

Reviewer 2 Report

In this prospective observational study, Riga et al investigated pro-inflammatory IL-8 serum concentration in patients with different origin of IAI in perioperative period. The study showed the level of serum IL-8 in patients with intra-abdominal infections of different origins is lower in comparison with healthy individuals, which is in contrast to previous reports. This work is to look into contradictory conclusion regarding IL-8 in inflammatory condition, but it is not well designed to resolve the controversy. 

- the sample size was not sufficient. 

- information of healthy controls was missed

- patients were not stratified in details (based on pathogen)

Author Response

Dear Reviewer,

many thanks for the constructive comments that improved the manuscript!The answers to your questions are in the attachment.

With best wishes,

Authors

Reviewer 3 Report

In the paper entitled “Serum Interleukin-8 in patients with different origin of intra-abdominal infections in perioperative period” by Dr. A. Riga et al, the authors evaluated the IL-8 serum amounts in patients subjected to surgery in the presence of intra-abdominal infection from different origins. Although the idea is appealing, there are several major issues and few minor issues.

Major issues:

1)     Demographic data in Table 1 for control group are missing.

2)     The study seems not well designed. Indeed, if patients with IAI were subjected to surgery, the real control group should be “relatively healthy” (without IAI) subjects subjected to surgery and with blood samples withdrawn at the same time points. In fact, the control group with baseline characteristics can be compared only with baseline characteristics of diseased patients.

3)     The manuscript is flawed by the use of inappropriate statistical analyses. Indeed, in order to detect differences between the group with serial blood sample a repeated measure two-way ANOVA or a non-parametric equivalent (although scarce, there are few options). This is because the data you collected violate the assumption of independence of observations which is a basic pre-requisite for all the commonly used statistical test (due to the high covariance correlation).

4)     The data presented in the three figures should instead be presented in the same figure (without the single points for all the patients), in order to appreciate the possible different trend of the three groups during time.

5)     Although the authors claims that they performed a ROC curve analysis, no data (image of the curve) are presented. In addition, how was the cut-off determined for ROC curve? Usually the Youden’s index is preferred for the determination of the best cut-off giving the most significant sensitivity/specificity parameters.

6)     How do they comment on the presence of higher levels of IL-8 in healthy controls compared to baseline IAI patients? It quite counterintuitive since the patients should be inflamed.

7)     The temperature of conservation of specimens (-20°C) is not appropriate for the analyses of cytokines since degradation during long term storage is likely to occur: -40°C or at best -80°C are mandatory. Where control subjects more “fresh” than the other patients? This might explain the different levels of IL-8.

8)     There is not any disclosure about the limitations of the study in the discussion section, which is almost mandatory for clinical studies.

Minor issues:

1)     The manuscript needs extensive language editing from a Native English, since it is hard to follow. Moreover, check for truncated sentences.

Author Response

Dear Reviewer,
we have included all your comments in the reviewed manuscript. For statistics, we present the originals.
We apologize, many questions were not clear. Your comments helped us formulate the limitations of the study.
The answers to your questions are in the attachment.
Still, we would like to highlight the strengths of our research.
1 At first time serum pro-inflammatory IL-8 was assessed in the East Ukrainian adults with Intra-abdominal infections, especially, postoperative (nosocomial) abscesses of the abdominal cavity.
2. The diagnostic role of serum IL-8 in the formation of postoperative abscesses has been shown, which requires an expansion of the sample and stratification of patients.
3. It has been shown of trajectory of change of IL-8 depending on the localization of inflammation at the peak of a localized disease, when surgical intervention is required, i.e. critical condition of the patient, and its dynamics after intervention in the early postoperative period. It has been shown on the example of patients with diffuse peritonitis, who received only laparotomic access, an excellent trajectory of changes of IL-8 due to a more extensive surgical trauma and less localized inflammation.
4. The determination of the fall of IL-8 in the late postoperative period may serve as an empirically early marker of abscess formation even before it is diagnosed with ultrasound and CT because the nosocomial infections are more severe with higher mortality
5. For further research is important of the accumulation of data and the study of pro-inflammatory IL-8 depending on the microbial agent, depending on the localization of abscesses, depending on the multiplicity of abscesses, depending on comorbide conditions, in particular, diabetes, depending on gender, depending on the level of IL-8 in the contents of the abscess, depending on the generalization of the infectious process and on the antibacterial drug taken. Research should be carried out on various population groups. Perhaps this will contribute to new methods of not only diagnosing and predicting, but also
Our study is the foundation for which all later studies depend on. They can provide the groundwork for what worked and what didn't work, what are the useful stratification factors, and it can be the early indicator useful. Finally it provide for determining the number of issues needed for a full study.
Many thanks again!
Authors

Reviewer 4 Report

Manuscript details:
Journal: Medical Sciences
Manuscript ID: medsci-550670
Type of manuscript: Article
Title: Serum Interleukin-8 in Patients with Different Origin of Intra-abdominal Infections in Perioperative Period  

Review

The study demonstrates a changes in the concentration of the serum proinflammatory IL-8 in patients with various origins of intra-abdominal infections in the perioperative period.

The 56 cases included in the analysis of the presented manuscript are a relatively small number for good statistics and conclusions.

The authors are encouraged to describe the limitations of the presented study.

The manuscript requires careful English proofreading. Sometimes sentences lose meaning (grammar and typos).

Most of the comments were written in the pdf text.

Author Response

Dear Reviewer,

we have included all your comments in the reviewed manuscript. Thank you for the correction of the text and comments that allowed us to formulate the limitations of the study and improved our manuscript. We apologize, many questions were not clear to the reader. We have corrected the manuscript in English also. All answers to your comments in the attachment.

With best wishes,

Authors

Round 2

Reviewer 2 Report

The authors have addressed my concerns and the revised manuscript is improved. English language editing is needed. 

Author Response

Dear reviewerБ цe thank you for all the comments. We tried to сщщкусе everything that you recommended to us. We sent the manuscript on August 17 to English Editor MDPI on 17 Aug and are waiting now Could you be so very kind to give a direction on how to improve the manuscript. You have written in all our sections that they should be improved. Please specify what exactly. We will be very grateful to you. With best wishes, authors  

Reviewer 3 Report

RESPONSE TO POINT 1: OK.

RESPONSE TO POINT 2: In response 2, point 4, how could you exclude IL-8 changes due to “surgical trauma” and healing? You should at least mention it in the manuscript. My point is that if you are trying to use IL-8 as a diagnostic tool, you need a control population. The control population should be something very similar to your diseased population but without the event (so the intra-abdominal infection). Thus, the most appropriate control population should be one subjected to surgery but without infection. I see that there are several problems in its selection, mainly due to the possible influence of the underlying pathology on the levels of IL-8. The authors should at least acknowledge it.

RESPONSE TO POINT 3: The analysis the colleagues used in the manuscript you cited are indeed correct since it was not a longitudinal study. As I already pointed out in the previous response, the “issue” with longitudinal studies is that you cannot use the “conventional” statistical instruments (so simple ANOVA, Kruskall-wallis, Mann-Whitney or t-test) to detect differences since the data coming from the same individuals are highly correlated and therefore introduce a bias in the analysis. The only way to detect if your event can have influence on the levels of IL-8 is though a repeated-measures approach which analyses all the data you’ve got and, more importantly, provides the so called “interaction” term ‘time*group’. This term is what you want to observe to have a clue on the possible effect of the different types of intra-abdominal infections (group) over time on IL-8. The “a posteriori” calculation of power of a test is rarely used as a good tool for detect an effect or to identify if it is significant. So, my suggestion is to repeat the statistical analyses by using another test (for instance a two-way ANOVA).

RESPONSE TO POINT 4: The answer is ok. If possible, they could add the graphs as supplementary material online.

RESPONSE TO POINT 5: The answer is ok. If the authors don’t want to add the figures to the final version of the manuscript, the should be included in some supplementary material online since I found them very informative.

RESPONSE TO POINT 6: OK.

RESPONSE TO POINT 7: OK.

RESPONSE TO POINT 8: OK.

RESPONSE TO POINT 9: OK.

Author Response

Dear Reviewer, we thank you for all the comments! We tried tocorrect everything that you recommended to us. We sent the manuscript on August 17 to correct English for MDPI English Edition. We conducted ANOVA, it was interesting and visiable. We hope to receive the manuscript from the translation and upload the already corrected text. We will ask the supportive team to place all additional figures on website. We have loaded more detailed comments in the response to the reviewer along with the results of the statistical analysis With best wishes, authors  
